# Comparative Analysis of Transcriptome Data of Wings from Different Developmental Stages of the *Gynaephora qinghaiensis*

**DOI:** 10.3390/ijms26083562

**Published:** 2025-04-10

**Authors:** Guixiang Kou, Yuantao Zhou, Haibing Han, Zhanling Liu, Youpeng Lai, Shujing Gao

**Affiliations:** 1Grassland Research Institute, Chinese Academy of Agricultural Sciences, Hohhot 010010, China; kgx3272647933@163.com (G.K.); hhb.25@163.com (H.H.); 2Institute of Plant Protection, Qinghai Academy of Agriculture and Forestry, Xining 810016, China; 3College of Agriculture and Animal Husbandry, Qinghai University, Xining 810016, China; zhouyt@qhu.edu.cn (Y.Z.); liu20000607ling@163.com (Z.L.)

**Keywords:** *Gynaephora qinghaiensis*, lepidoptera, wing development, differently expressed genes

## Abstract

*Gynaephora qinghaiensis* is a major pest in the alpine meadow regions of China. While the females are unable to fly, the males can fly and cause widespread damage. The aim of this study was to use transcriptome analysis to identify and verify genes expressed at different developmental stages of *Gynaephora qinghaiensis*, with particular emphasis on genes associated with wing development. High-throughput sequencing was performed on an Illumina HiSeqTM2000 platform to assess transcriptomic differences in the wings of male and female pupa and male and female adults of *Gynaephora qinghaiensis*, and the expression levels of the differentially expressed genes (DEGs) were verified by real-time fluorescence quantitative PCR (RT-qPCR). A total of 60,536 unigenes were identified from the transcriptome data, and 25,162 unigenes were obtained from a comparison with four major databases. Further analysis identified 18 DEGs associated with wing development in *Gynaephora qinghaiensis*. RT-qPCR verification of the expression levels showed consistency with the RNA sequencing results. Spatio-temporal expression profiling of the 18 genes indicated different levels of expression in the thoraces of male and female pupa, as well as between the wing buds of adult females and the wings of adult males. GO annotation analysis showed that the DEGs were associated with similar categories with no significant enrichment and were involved in cellular processes, cellular anatomical entities, and binding. KEGG analysis indicated that the DEGs were associated with endocytosis and metabolic pathways. The results of this study expand the information on genes associated with *Gynaephora qinghaiensis* wing development and provide support for further investigations of wing development at the molecular level.

## 1. Introduction

*Gynaephora qinghaiensis* (Lymantriidae, Lepidoptera) is commonly known as the red-headed black caterpillar [1]. To date, a total of 15 species have been identified throughout the world, with 13 species in Asia, of which 8 are endemic to Tibetan Plateau in China [2,3]. *Gynaephora qinghaiensis* is distributed primarily in the alpine meadows of Haiyan, Tianjun, Zeku, Mado, Qilian, and Menyuan. These alpine meadows of the Qinghai–Tibetan Plateau are rich in plants belonging to the Cyperaceae, Carex myosuroides Vill., and Poaceae Barnhart families, all of which are important for the growth and development of *G. qinghaiensis* [4,5,6]. The larvae have poisonous glands on their backs, presenting a danger to humans and animals. Additionally, *G. qinghaiensis* is difficult to control and causes widespread damage to the ecology of the grasslands [7]. The morphology of the caterpillars differs markedly between males and females; females have degraded thoracic peduncles, wings, and antennae, while males are more migratory [8]. In addition, several studies have reported that *G. qinghaiensis* are highly adaptive and can tolerate harsh environmental conditions, including strong ultraviolet radiation, hypoxia, and severe cold [9]. Current research on these insects has focused on gene expression and analyses of high-altitude stresses [10], phylogeography of endemic grassland caterpillars on the Tibetan Plateau [11], and whole mitochondrial genome sequencing [12], among other areas. Little is known about its wing development.

The rapid development of high-throughput sequencing techniques has led to numerous transcriptomic investigations of non-model insects, with the transcriptome representing the collection of all mRNAs transcribed by a particular tissue or cell at a particular developmental stage or functional state [13]. It is thus possible to link genetic information with insect phenotypes under different conditions, contributing to the understanding of insect gene function, growth and development, phylogeny, physiology, and biochemistry [14]. The analysis of differentially expressed genes (DEGs) is inextricably linked to transcriptome sequencing. DEGs are generated by the interaction of various processes, including internal signaling, external conditions, genetic variation, DNA methylation, RNA splicing, and other mechanisms, which all affect the expression level of the same gene in different cells or tissues, enabling the development of different organisms, their ability to adapt to environmental changes, and the acquisition of specific functions. To date, numerous studies have used transcriptome sequencing to analyze differential gene expression in insects. For instance, screening the antenna transcriptomes of male and female *Glenea cantor* (Fabricius) beetles identified 488 DEGs, of which 268 were upregulated and 220 were downregulated [15], while 268 DEGs were found between the tentacles of male and female *Callosobruchus maculatus* (Fabricius) [16]. Yuan et al. used both transcriptomic analysis and 1H NMR-based metabolomics to explore the potential mechanisms associated with flupyrimin resistance in *Spodoptera litura* [17], while transcriptome analysis of *Bradysia odoriphaga* by Chen et al. identified four P450 genes related to imidacloprid metabolism [18]. It is thus apparent that transcriptomics is increasingly widely used. Transcriptomics would thus be a useful means of investigating the mechanisms underlying insect wing development.

Insect wings are important for flight, adaptation to the environment, and reproductive survival, as well as in warning and avoiding natural enemies. The study of insect wings also contributes to many fields of study, including evolutionary biology, ecology, and physiology [19]. Studies on insect wings have focused on the insects belonging to the orders Coleoptera, Lepidoptera, and Hemiptera; Zhang et al. studied the ultrastructure of four insect sheath wing sections to provide a basis for bionic design, while Wu et al. found that the fore and hind wings of *Sogatella furcifera* have a wing surface structure with a thin center and thick edges, as well as small uniformly distributed spiny protrusions and multiple sensors on the backs of the forewings, contributing to an understanding of the insect’s migration and dispersal, and suggesting ideas for their prevention and control, and Hong et al. reported that the scales of four species of tigonopteryx butterflies did not differ significantly in terms of ultrastructure [20,21,22]. In recent years, there have been reports on the current status of research on wing development at the molecular level. Lu et al. successfully characterized the AcBurs-α and AcBurs-β genes in the pest, *Aphis citricidus*, and concluded that the AcBursicon gene plays a crucial role in *Aphis citricidus* wing development, as well as representing a potential molecular target for the control of the pest using RNAi-based approaches [23]. Zhang et al. showed that CYP311A1 is closely involved in regulating lipid metabolism and morphogenesis in *Drosophila* wings [24], while Polajnar et al. found that *Psylla chinensis* communicates between the sexes through signals generated by wing vibrations and that these had specific spectral structures which may be important for interspecific recognition. This finding not only reveals a new mechanism of sound production in this species but also provides important information for the development of control strategies [25]. Yu et al. revealed a novel mechanism for silencing the Yki gene at the post-transcriptional level by miR-927, suggesting new perspectives for the management of pests [26]. Shang et al., in an investigation on *Toxoptera citricida* and *Acyrthosiphon pisum*, found that miR-9b targets and regulates the ATP-binding transporter protein ABCG4 and that inhibition of miR-9b expression at low insect population densities significantly enhanced the winging rate of the progeny [27]. Broehan et al. showed that ABC transporter proteins affect *Tribolium castaneum* wing development by regulating lipid synthesis and metabolism [28].

Due to its status as a major pest in alpine meadows, it is important to investigate the molecular mechanisms underlying wing development in *G.qinghaiensis* to develop environmentally friendly and sustainable development of affected grasslands. In this study, we performed high-throughput sequencing of transcriptome libraries from the wings of male and female pupa and adult males and females of *G. qinghaiensis* to identify genes associated with wing development and lay a foundation for further research on the mechanisms of wing development and flight in this species, together with targets for its control.

## 2. Results

### 2.1. Transcriptome Sequencing of Gynaephora qinghaiensis

An Illumina HiSeqTM2000 high-throughput sequencing platform was used for sequencing the transcriptomes of 12 samples from two different developmental stages of *G. qinghaiensis* (female pupa, female wing buds, male pupa, and male wings) and to complete the gene structure analysis based on the Unigenes library. After filtering, the sequences obtained for female pupa, female wing buds, male pupa, and male wings of *G. qinghaiensis* ranged from 36,205,752 to 57,571,540. We compared the reads obtained from sequencing with the Unigenes library, and the results showed that the mapping rates were all above 80%. After filtering the data, the quality of the data was assessed in terms of the composition and quality distribution of the bases. A total of 80.24 Gb of clean data was obtained from the sequencing, with the clean data of each sample reaching 5.36 Gb. After de novo assembly, a total of 60,536 Unigenes were obtained, with a GC content of 37.3% and above, while the number of N50s was 9342, representing 1732 bp in length, with an average length of 1011 bp. The raw data obtained from female pupa, female wing buds, male pupa, and male wings were 50,074,028 (FPW1, FPW2, and FPW3 for mean), 44,435,128 (FW1, FW2, and FW3 for mean), 42,284,808 (MPW1, MPW2, and MPW3 for mean), and 49,965,396 (MW1, MW2, and MW3 averaged), respectively. The percentage of bases in Q20 and Q30 were 97.66% and 94.48% and above, respectively (Table 1).

### 2.2. Functional Annotation of Unigenes from Gynaephora qinghaiensis

Using E-values not greater than 10^−5^ in BLAST1.4.0 (https://www.ncbi.nlm.nih.gov/), the unigene sequences were compared with the protein databases KOG, KEGG, Nr, and Swiss-Prot to identify proteins with the highest sequence similarities. This yielded annotations for 25,162 unigenes. The annotation statistics for each database were 11,335 annotations in KOG, 21,015 in KEGG, 24,577 in Nr, and 12,128 in Swiss-Prot (Figure 1).

### 2.3. Numbers of DEGs Associated with Wing Development in Gynaephora qinghaiensis

A total of 477 DEGs were found between female and male pupa in two different wing developmental stages (male and female pupa and male and female wings) of *Gynaephora qinghaiensis*, of which 34.8% (166) were upregulated and 65.2% (311) were downregulated. A total of 7527 DEGs were identified between female wing buds and male wings, of which 61% (4592) and 39% (2935) were upregulated and downregulated, respectively. A comparison of female pupa and female wings showed that 2864 genes were differentially expressed, with 51.5% (1474) and 48.5% (1390) upregulated and downregulated, respectively. Differential expression of 3305 genes was found between male pupa and male wings, of which 47% (1551) were upregulated and 53.1% (1754) were downregulated (Table 2).

### 2.4. GO Enrichment Analysis of DEGs in Gynaephora qinghaiensis

GO functional enrichment analysis of DEGs in the FPW vs. MPW comparison group showed that 477 DEGs were significantly enriched in 18 functional entries (*p* < 0.05). Eight entries were found in the biological process category, of which the most significant enrichments were seen in chitin metabolic processes, glucosamine-containing compound metabolic process, and amino sugar metabolic processes. Nine entries were found in the cellular component category, with the most significant being extracellular region, intrinsic component of plasma membrane, and the extracellular matrix, while the five entries in the molecular function category were associated with structural constituent of cuticle, structural constituent of chitin-based cuticle, and structural molecule activity (Figure 2A). GO functional enrichment analysis of DEGs in the FPW vs FW comparison group showed that 2864 DEGs were significantly enriched in 102 functional entries (*p* < 0.05), with the biological process category including 8 entries of which the most significant were chitin-based cuticle development, cuticle development, and lymphocyte anergy, and the cellular component category included 5 entries, with the most significant enrichment in extracellular region, extracellular matrix, and intrinsic component of plasma membrane, while the molecular function category had 30 entries with the most significant associated with the extracellular region, intrinsic component of plasma membrane, and the extracellular matrix (Figure 2B). GO functional enrichment analysis of DEGs in the FW vs. MW comparison group showed that 7527 DEGs were significantly enriched in 489 functional entries (*p* < 0.05). Biological process included 374 entries, and the most significantly enriched GO entries were found to be anatomical structure development, anatomical structure morphogenesis, and multicellular organismal process. The cellular component category had 4 entries, with the most significant being cellular anatomical entity, intracellular anatomical structure, and cytoplasm, while the molecular function classification included 111 entries, with the most significantly enriched being protein binding, binding and xenobiotic transmembrane, and binding and xenobiotic transmembrane transporter activity (Figure 2C). GO functional enrichment analysis of DEGs in the MPW vs. MW comparison group showed that 3305 DEGs were significantly enriched to 213 functional entries (*p* < 0.05). Of these, 213 were in the biological process category, with the most significant involving mitochondrial translational termination, mitochondrial translational elongation, and mitochondrial translation cells. There were 40 entries in the cellular component category, with the most significantly enriched being mitochondrial protein-containing complex, mitochondrial ribosome, and organelle inner membrane, while 22 entries were found in molecular functions, most of which were linked to structural components with the most significant enrichment in structural constituent of cuticle, structural constituent of chitin-based larval cuticle, and structural constituent of chitin-based cuticle (Figure 2D).

### 2.5. KEGG Enrichment of Gynaephora qinghaiensis DEGs

The 3520 DEGs in the FPW vs. MPW comparison group were found to be enriched in 62 metabolic pathways, the most significant of which (*p* < 0.05) being metabolic pathways, fatty acid metabolism, fatty acid elongation, fatty acid degradation, thiamine metabolism, and the citrate cycle (TCA cycle) (Figure 3A). In the FW vs. MW group, the 7527 DEGs were enriched in 136 metabolic pathways, of which 14 were significant (*p* < 0.05), including inositol phosphate metabolism, fatty acid degradation, fatty acid elongation, and endocytosis (Figure 3B). The 2864 DEGs in the FPW vs. FW comparison group were enriched in 121 metabolic pathways, of which the 19 most significant (*p* < 0.05) included metabolic pathways, insect hormone biosynthesis, metabolism of cytochrome P450 to xenobiotics, biosynthesis of unsaturated fatty acids, ascorbate and aldehyde metabolism, metabolism of amino acids and nucleotides, drug metabolism—cytochrome P450, drug metabolism—cytochromes, drug metabolism—other enzymes, fatty acid metabolism, and ABC transporters (Figure 3C). The 3305 DEGs in the MPW vs. MW group were enriched in 126 metabolic pathways, of which the 9 most significant (*p* < 0.05) were oxidative phosphorylation and metabolic pathways (Figure 3D).

### 2.6. Validation and Expression Profiling of DEGs for Wing Development in the Gynaephora qinghaiensis

Verification of the expression of 18 DEGs associated with wing development using RT-qPCR yielded results consistent with the changes observed in the RNA-Seq data, confirming the reliability of the RNA sequencing (Figure 4). The expression levels of genes associated with wing development in different tissues of *G. qinghaiensis* were analyzed by RT-qPCR, using female pupa thorax tissue as the control. It was found that the 18 genes showed different levels of expression in the thoraces of male and female pupa, in the wing buds of adult females and the wings of adult males. Seventeen development-associated genes were found to be expressed in the wing buds of adult females, with the exception of the *Vg* gene, and the expression of *UGT-3* was two-fold higher than that of *RPS6-1*. Among them, the expression of *UGT-1*, *GST-1*, *GST-5*, *GST-6*, *RPS6-2*, *Ecr*, *Pached*, *Ubx*, and *Burs* was significantly higher in the thorax of female pupa than the other nine genes (*p* < 0.05). The expression of *GST-1*, *GST-5*, and *GST-6* in the thorax of male pupa was significantly higher than that in the thorax of female pupa (control group) (*p* < 0.05), and the expression of the other 15 genes in the thorax of male pupa was lower than that in the thorax of female pupa (control). The expression of the remaining 15 genes in the thoraces of male pupa was lower than that in the thoraces of female pupa (control). The genes showing significant differences between adult male wings and female pupa thoraces (control) (*p* < 0.05) were *UGT-3*, *GST-2*, *GST-3*, *GST-6*, and *RPS6-1*, while those with expression less than 0.5 were *UGT-1*, *GST-6*, *Vg*, *Srf*, and *Burs* (Figure 5).

## 3. Discussion

In this study, we performed transcriptome sequencing and analysis of differentially expressed genes in the wings of male and female pupa and adults of *Gynaephora qinghaiensis* on an Illumina Hiseq 2000 platform, identifying genes associated with wing development. The results lay a foundation for subsequent research on the molecular mechanisms responsible for insect flight. A total of 60,536 unigenes were attained from the transcriptome sequencing data, with an N50 length of 1732 bp and an average length of 1011 bp, indicating that the integrity of the sequences obtained from the assembly was high [29]. The percentages of Q20 and Q30 bases were 97.66% and 94.48% and above, respectively, with reliable sequencing quality [30]. Therefore, the assembly quality and length of the sequencing data met the basic requirements for further analysis of the transcriptome and identification of differentially expressed genes.

KEGG enrichment analyses indicated that the DEGs in the FPW vs. MPW comparison group were enriched primarily in metabolic pathways and fatty acid metabolism pathways, suggesting that the metabolic levels of female and male pupa of *G. qinghaiensis* differed in their fat contents. The functions of the DEGs in the FW vs. MW group were mainly associated with inositol phosphate metabolism, suggesting that stress tolerance functions differed between FW and MW. The functions of the DEGs in the FPW vs. FW group were significantly enriched in metabolic pathways and insect hormone biosynthesis, suggesting that the hormone levels changed during the pupa instars of *Gynaephora qinghaiensis*. The functions of the DEGs in the MPW and MW comparison were significantly enriched in oxidative phosphorylation and metabolic pathways, and we therefore hypothesized that changes in energy storage and release, protein degradation, and other pathways in the prairie caterpillar during male metamorphosis from pupa to adult promote biological processes involved in plumage. Eighteen genes associated with wing development were identified, namely, *UGT-1*, *UGT-2*, *UGT-3*, *Ecr*, *GST-1*, *GST-2*, *GST-3*, *GST-4*, *GST-5*, *GST-6*, *Ubx*, *Srf*, *Burs*, *Pached*, *Vg*, *Rps6-1*, *Rps6-2*, and *Fringe*, and their expression was verified by RT-qPCR to be consistent with the RNA sequencing results. *UGT* is involved in glycosylation reactions, specifically in toxin degradation reactions, which are essential for the elimination and inactivation of endogenous or exogenous toxic substances in insects [31]. This was demonstrated in *Spodoptera frugiperda* in vivo by silencing specific UDP-glycosyltransferases using SfUGT33F28 gene-specific dsRNA to inactivate maize defenses 3,4-dihydro-2H-1,4-benzoxazin-6-ol analogs [32]. The expression of UGT-3 was found to be two-fold higher than that of RPS6-1. *GSTs* play an important role in the detoxification of heterologous compounds and have been linked to insecticide resistance [33]. These enzymes primarily catalyze the binding of electrophilic compounds to the sulfhydryl group of reduced glutathione (GST), leading to nucleophilic attack on the substrate, resulting in increased solubility of the product and enabling excretion [34]. It was found that elevated *GSTs* activity could accelerate the metabolic processing of exogenous toxic substances and enhance insect adaptation to adverse conditions [35]. The expression of *GSTs* in the thoraces of male pupa differed significantly from that in the thoraces of female pupa (control group). Therefore, it was further hypothesized that the promotion of glycosylation by *UGT* enzymes and the catalytic functions of GSTs in *Gynaephora qinghaiensis* enables the degradation of toxic substances in the food, preventing adverse effects on wing development. Burs are hormone-like proteins that regulate epidermal hardening and wing span in insects [23]. A preliminary study found that the expression of the ellagitannin gene in winged adult aphids was higher than that in both wakame and wingless adult aphids, and disruption of ellagitannin gene expression resulted in failure of wing development in *Myzus persicae* [36]. Wu et al. found that the wings of *Laodelphax striatellu* appeared mutilated, the anterior and posterior wings did not extend normally, and the wing surfaces did not darken normally after injection of dsRNA specific for the *EcR* gene into the 4th instar wakame of *Laodelphax striatellu* [37]. In the present study, the expression of the *Burs* gene differed between female pupa and adult females and female and male pupa, while that of *EcR* was upregulated in all four groups, so we hypothesized that *Burs* and *EcR* have key roles in regulating wing extension and epidermal coloration in the pupa stage of *Gynaephora qinghaiensis*.

*Ultrabithorax* belongs to the BX-C gene cluster [38], and its expression is important for the formation of the insect body axis, the development of the nervous system, and the genesis, growth, and differentiation of tissues and organs [39]. *Ubx* genes are expressed primarily in the posterior thoracic to abdominal regions of insects. In *Drosophila*, the expression is mainly concentrated in the 2nd thoracic to 8th abdominal segments [40], while in Bombyx mori, it extends from the 2nd thoracic to the 9th abdominal segments [41]. Usually, the expression levels of *Ubx* in the 3rd thoracic and 1st abdominal segments are significantly higher than those in other parts of the body, suggesting that the principal site of action of *Ubx* may be in this region during insect development. Numerous studies have shown that *Ubx* is primarily involved in the differentiation and development of appendages such as wings and legs [39]. In studies on *Bombyx mori* and Tribolium castaneum, it was found that *Ubx* was also key to regulating the morphological differentiation of the fore and hind wings—but the mode of regulation mediated by *Ubx* and its target genes varied in different species—and that, in addition to regulating the differentiation of the fore and hind wings, *Ubx* was also involved in the development of the flight muscles [42]. Our experimental results showed that *Ubx* expression was downregulated between female pupa and adult females, while showing upregulation in the other three groups. It is thus hypothesized that this gene is involved in the development of the anterior and posterior wings of male adults of *Gynaephora qinghaiensis*, as well as of their flight muscles, which drive the movement of the anterior and posterior wings, thus achieving long-distance flight.

*Srf* is an ancient member of the MADS box family and has been strongly conserved during evolution [43]. *Srf* genes have important roles in *Drosophila* wing regulation [44], tracheal cell differentiation [45,46], cell migration, and actin-regulated processes [47,48]. During the development of wing veins and venation in the *Drosophila* wing, the primary function of *DSRF* is controlling the terminal differentiation of venation cells, thereby promoting the formation of venation, as well as controlling the extent and location of the wing veins. Lyulcheva et al. found that *Srf* promotes cell proliferation in *Drosophila* [49], while a study by Thompson suggested that it is the regulation of the induced cell that makes it functional [50]. An absence of the *DSRF* gene results in ectopic wing vein expression at the site of the vein. Conversely, it can also result in the absence of the wing vein, and the regulation of wing vein and vein formation by *DSRF* is recognized as a stoichiometric effect [44,51]. In our experiments, *Srf* gene expression was downregulated between female pupa and adult females and between female and male pupa and was upregulated between female and male wings and male pupa and male wings. The Srf gene may play a key role in the regulation of the proliferation or induction of wing development cells during the wing development of adult *Gynaephora qinghaiensis*; there are many dense scaly hairs on the wings of males, which is under the control of the Srf which regulates the growth and development of hairs on the wing surfaces.

*Vg* is an evolutionarily conserved gene involved in wing development in insects that triggers a regulatory network of *Vg* expression and wing development through proliferation and cell aggregation [52,53]. During the formation of the wing primordium, the dorsal cells secrete the morphogenetic protein Decapentaplegic (Dpp), and in the region close to the dorsal cells of the insect, the Distal-less (Dll) gene is stimulated to express Vg in response to a high concentration of Dpp, and these Vg protein-containing cells are gradually transferred to the dorsum to form the wing primordium [54,55]. Expression profiling showed that the Vg gene was not expressed in adult female wing buds, and we hypothesize that the Vg gene also has similar functions in the development of wing buds of adult *Gynaephora qinghaiensis* females.

*Rps6* is a key structural protein making up the 40S small subunit of the ribosome and is the first ribosomal protein to be phosphorylated [56]. It has been shown that Rps6 binds to a class of mRNAs encoding ribosomal proteins and translation initiation factors mainly in the phospho-RPS6 phosphorylated form, thereby regulating a variety of complex intracellular gene translation processes that ultimately regulate the cellular growth cycle, as well as influencing biological activities such as protein synthesis, cell growth, and proliferation [57]. In our study, *Rps6-1* was downregulated between female pupa and adult females and upregulated in the other three groups, while Rps6-2 was upregulated in all four groups. Therefore, it is hypothesized that *Rps6-1* and *Rps6-2* are able to promote protein synthesis, cell growth, and proliferation during wing development in *Gynaephora qinghaiensis*.

Signaling regulation of Fringe proteins is transduced through the Notch receptor [58], and binding of the Notch, extracellular EGFR-bound, O-conjugated fucose to the N-ethylphthalimidoglucan moiety prevents ligand-induced proteolytic cleavage and inhibits Notch activation [59]. In *Drosophila*, the Notch signaling pathway regulates the initiation of cell differentiation and also plays an important role in organogenesis and morphogenesis during embryonic development [60]. Notch can also control normal wing development [61]. It has been shown that the deletion of *Fringe* can lead to abnormal wing boundary development in *Drosophila* [62]. Fringe genes in this experiment were downregulated between female pupa and adult females and female and male pupa and were upregulated between female and male wings and male pupa and male wings. We hypothesize that Fringe interacts with the Notch receptor during adult wing development in *Gynaephora qinghaiensis*, regulating cell differentiation and thus controlling normal wing development.

## 4. Materials and Methods

### 4.1. Insects Materials and Sampling

The *Gynaephora qinghaiensis* samples used in this study were collected from Haiyan County, Haibei Prefecture, Qinghai Province (109°99′ E, 36°89′ N) in July 2023. Larvae were collected in an alpine meadow grassland and were transferred to the laboratory for rearing, and samples of *G. qinghaiensis* in two different developmental stages (pupal stage and adult stage) were selected for investigation. The thoraces of insects (male and female) in the pupa stage, the wing bud portion of adult females, and the wing base of adult males were dissected into centrifuge tubes which were carefully placed in a foam box filled with liquid nitrogen to prevent degradation of the sample. Two different developmental periods (four different tissues) of *G. qinghaiensis*, male and female pupa thoraces and adult female wing buds and adult male wing bases, were selected for transcriptome sequencing, with three sets of replicates per sample. The samples were termed FPW for female pupa thorax, MPW for male pupa thorax, FW for adult female wing buds, and MW for adult male wing bases.

### 4.2. Total RNA Extraction of Trichoderma reesei from Gynaephora qinghaiensis

Samples of male and female pupa and male and female wings of *G. qinghaiensis* were tested and qualified for library construction. After enrichment of mRNAs with polyA tails by magnetic beads with Oligo(dT), the mRNAs were fragmented ultrasonically. The fragmented mRNA was used as a template with random oligonucleotides as primers to synthesize first-strand cDNA using a M-MuLV reverse transcriptase system, followed by degradation of the RNA strand with RNaseH and synthesis of the second strand of cDNA with dNTPs in the DNA polymerase I system. The purified double-stranded cDNA was end-repaired, A-tailed, and ligated to the sequencing connector, with cDNAs around 200 bp identified using AMPure XP beads. The DNA was amplified by PCR, and the PCR products were re-purified using AMPure XP beads to obtain the libraries. To ensure sequencing quality, strict conditions were applied during library construction, by using agarose gel electrophoresis to analyze the integrity of the RNA and the presence of DNA contamination, a NanoPhotometer spectrophotometer (Implen, Munich, Germany) to detect the purity of the RNA (OD260/280 and OD260/230 ratios), a Qubit2.0 NanoPhotometer (Invitrogen, Carlsbad, CA, USA) (OD260/280 and OD260/230 ratio) to assess RNA purity, a Qubit2.0 Fluorometer (Invitrogen, Carlsbad, CA, USA) for quantification, and an Agilent 2100 (Agilent Technologies, Santa Clara, CA, USA) bioanalyzer for the assessment of integrity.

### 4.3. Identification of DEGs

The reads were assembled using Trinity v2.0.2 (The Broad Institute, Cambridge, MA, USA) software. Trinity first concatenated reads with a certain length overlap into longer fragments. These assembled fragments without N, obtained through the reads overlap relationship, were used as an assembled Unigene. The quality of the assembly results was assessed using the N50 value and sequence length, and the Benchmarking Universal Single-Copy Orthologs (BUSCO) (http://busco.ezlab.org/) (Access date: 10 December 2024) was evaluated. Unigene sequences were aligned to the protein databases NR, SwissProt, KEGG, and COG/KOG by blastx (Evalue < 0.00001) to obtain the protein with the highest sequence similarity for a given Unigene, which resulted in protein functional annotation information for that Unigene.

### 4.4. Screening and Functional Enrichment of DEGs

The reads obtained from the sequencing were compared with the Unigenes library, and based on the results of the comparison, the expression level was estimated using RSEM. Using the fragments per kilobase per million mapped fragments (FPKM) values corresponding to the expression of the unigenes, DESeq2 was used to identify DEGs using the criteria of fold change ≥ 2 and false discovery rate (FDR) < 0.05. The *G. qinghaiensis* data were compared with the Unigene sequences using BLAST (https://www.ncbi.nlm.nih.gov/) in the GO and KEGG databases, and after predicting the amino acid sequences of the Unigenes, HMMER website (http://www.ebi.ac.uk/Tools/hmmer/) was used to compare the sequences with the Pfam database for annotation.

### 4.5. RT-qPCR Verification of DEGs

DEGs were identified from the transcriptome data, and genes related to wing development in *G. qinghaiensis* were selected. These included UDP-glucuronosyl transferase (UGT-1, UGT-2, and UGT-3), Glutathione S-transferase (GST-1, GST-2, GST-3, GST-4, GST-5, and GST-6), Ecdysone receptor (Ecr), Ultrabithorax (Ubx), Serum response factor (Srf), Bursicon (Burs), Patched, Vestigial (Vg), Ribosomal protein S6 (Rps6-1, Rps6-2), and Fringe, while Rps15 was used as the internal reference gene. Primer 5 software was used to design 18 pairs of primers for qPCR; the primer sequences and related information are shown in Table 3. The qPCR amplification program was 95 °C for 3 min, followed by 35 cycles (95 °C for 30 s, 56 °C for 30 s, and 72 °C for 40 s). The 2^−ΔΔCt^ method was used to calculate the relative expression of the target genes (Table 3).

### 4.6. Data Processing

The relative expression of 18 DEGs associated with wing development in different tissues and developmental stages of the Qinghai grassland caterpillar was calculated using the 2^−∆∆Ct^ method, in which ∆Ct = Ct (target gene) − Ct (internal reference gene) and ∆∆Ct = ∆∆Ct (sample) − ∆∆Ct (control). Statistical analysis was performed using SPSS 26.0, using ANOVA for comparing the differences in the expression of these 18 genes in different tissues and developmental stages, as well as multiple comparisons using Duncan’s New Compound Polar Deviation method. The expression profiles were plotted using GraphPad Prism 9.5.

## 5. Conclusions

The *Gynaephora qinghaiensis* is one of the primary pests on the Qinghai–Tibet Plateau, posing a significant threat to the health of grassland ecosystems. Due to its unique sexual dimorphism mechanism, only male adults possess the ability to fly. Therefore, we aim to leverage research on wing development to implement control measures for the grassland caterpillar, thereby safeguarding the ecological security of the grasslands. This study analyzed wing development in *Gynaephora qinghaiensis* using transcriptome sequencing. The transcriptome data were analyzed by GO annotation and KEGG pathway enrichment, and differentially expressed genes were verified by real-time fluorescence quantitative PCR. It was discovered that 18 wing-development-related genes were involved in wing development in *Gynaephora qinghaiensis*. The results of the study not only enrich the genetic data on *Gynaephora qinghaiensis* at the transcriptional level but also lay the foundation for exploring the molecular mechanisms underlying the development of flight in this species, as well as provide a reference for the identification of molecular targets for the environmentally friendly control of *Gynaephora qinghaiensis*.

## Figures and Tables

**Figure 1 ijms-26-03562-f001:**
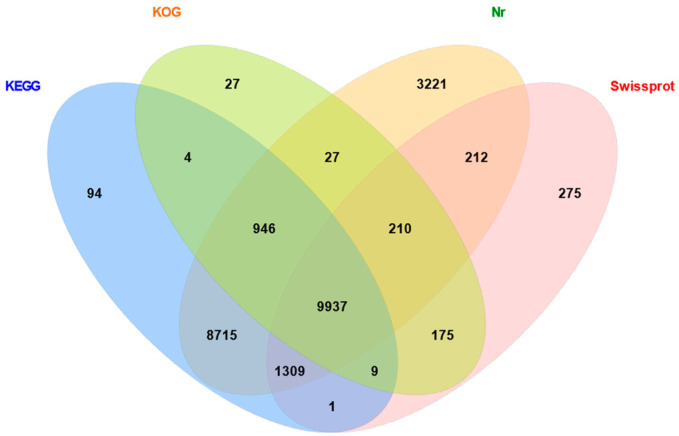
Wayne diagram showing annotations in four major databases for genes associated with wing developmental periods in *Gynaephora qinghaiensis*.

**Figure 2 ijms-26-03562-f002:**
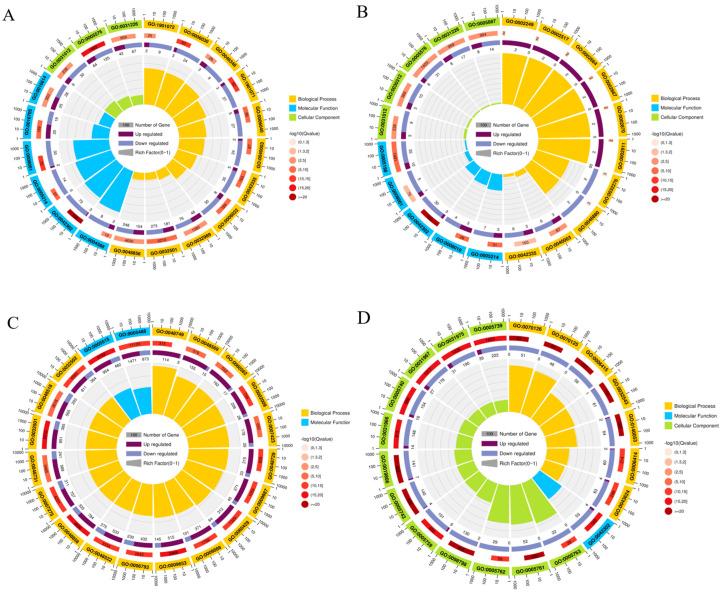
GO enrichment analysis of *Gynaephora qinghaiensis* at two different wing developmental periods. (**A**) GO enrichment for FPW vs. MPW. (**B**) GO enrichment for FPW vs. FW. (**C**) GO enrichment for FW vs. MW. (**D**) GO enrichment for MPW vs. MW. Yellow: biological process; Blue: molecular function; Green: cellular component; Purple: Up regulated; Haze blue: Down regulated; Dark gray: Number of Gene; Light gray: Rich factor; Light pink~Dark red: Correlation.

**Figure 3 ijms-26-03562-f003:**
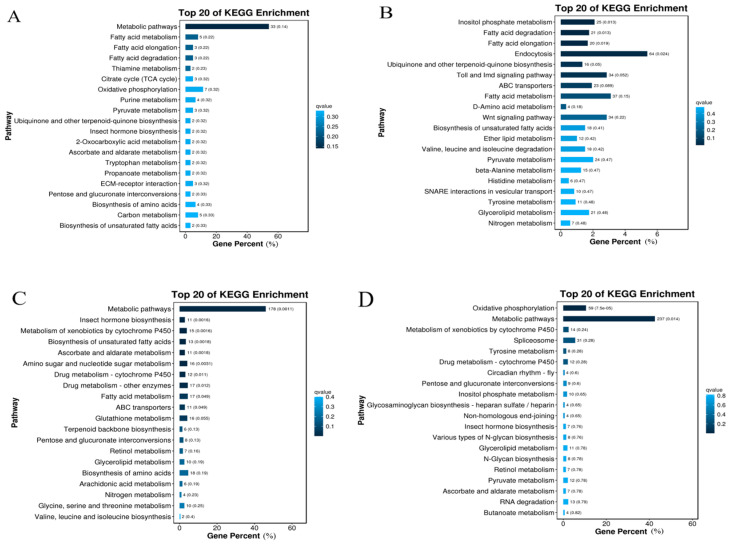
KEGG enrichment analysis of *Gynaephora qinghaiensis* at different wing developmental periods. (**A**) KEGG enrichment of FPW vs. MPW. (**B**) KEGG enrichment of FW vs. MW. (**C**) KEGG enrichment of FPW vs. FW. (**D**) KEGG enrichment of MPW vs. MW.

**Figure 4 ijms-26-03562-f004:**
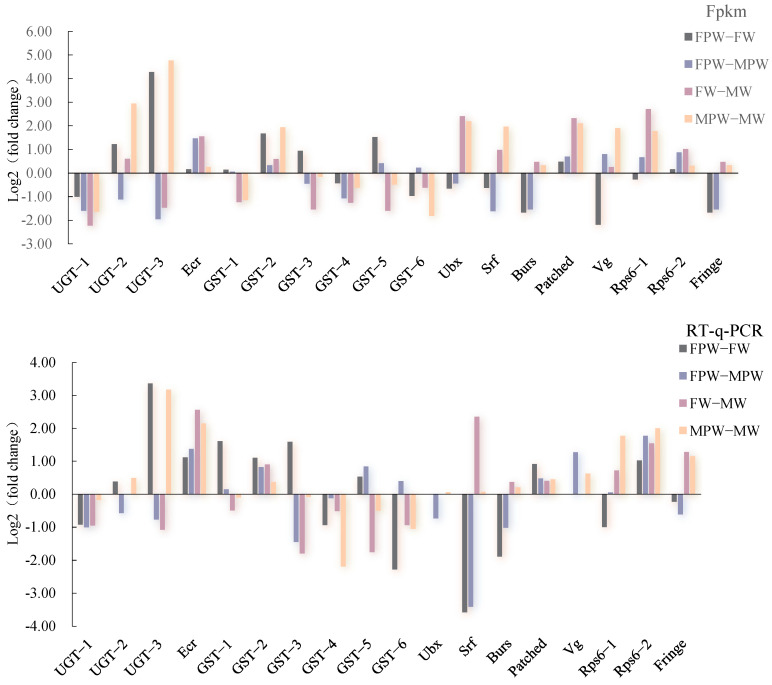
RT-qPCR verification of the RNA sequencing results of DEGs from two different wing developmental periods of *Gynaephora qinghaiensis*.

**Figure 5 ijms-26-03562-f005:**
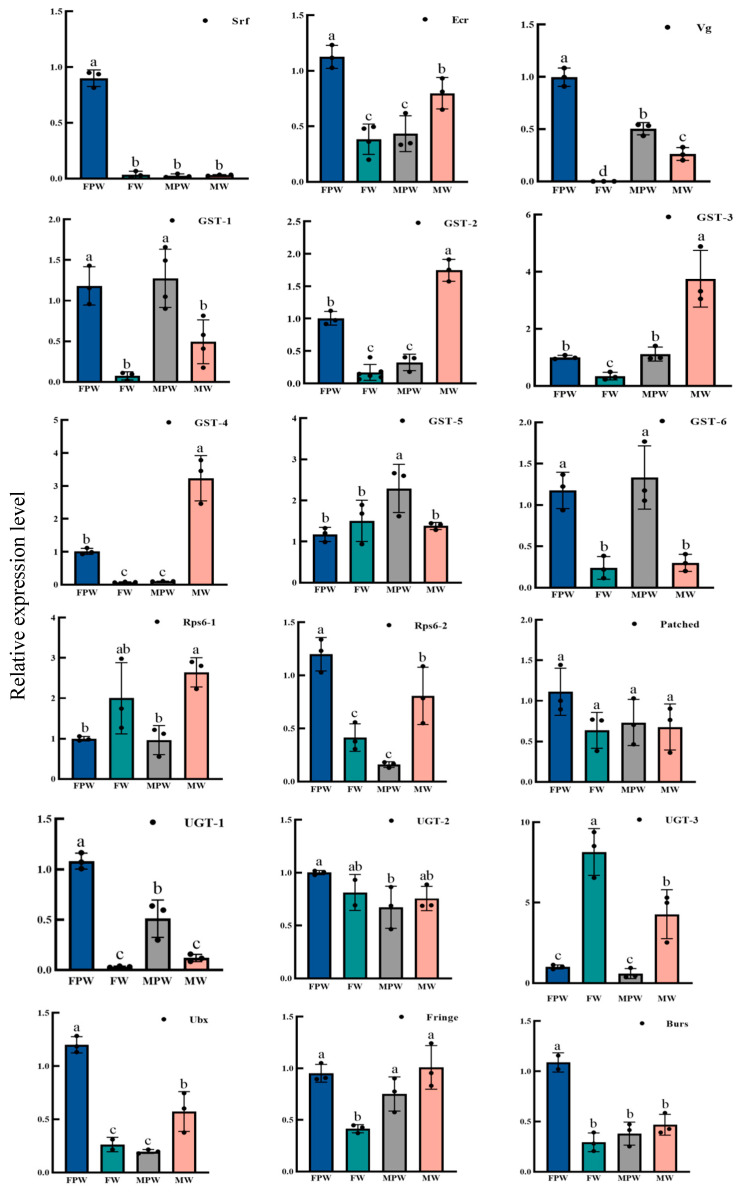
Tissue expression profiling of wing-development-associated genes in *Gynaephora qinghaiensis*. Vertical coordinates refer to relative expression; horizontal coordinates refer to different developmental periods (FPW: female pupa thorax; MPW: male pupa thorax; FW: adult female wing buds; MW: adult male wings). Significant level: a > b > c > d (*p* < 0.05).

**Table 1 ijms-26-03562-t001:** Statistics of the transcriptome data of wing development of *Gynaephora qinghaiensis* during different developmental periods.

Tissue Part	Sample ID	Clean Data/Article	Mapping Rate	Q20(%)	Q30(%)	Clean Data/bp
Female pupa	FPW1	40,171,912	84.41%	98.22%	96.59%	5,986,819,607
	FPW2	57,571,540	85.97%	97.66%	94.48%	8,583,049,456
	FPW3	51,834,670	84.96%	98.07%	96.36%	7,717,310,487
Female wing buds	FW1	46,961,368	84.81%	97.79%	95.16%	7,002,870,012
	FW2	42,698,294	83.52%	98.10%	96.42%	6,362,782,499
	FW3	36,469,654	83.35%	98.20%	96.54%	5,432,263,013
Male pupa	MPW1	45,752,948	86.25%	98.12%	96.45%	6,807,876,453
	MPW2	48,991,512	85.91%	98.15%	96.49%	7,298,753,786
	MPW3	38,033,428	84.32%	98.27%	96.74%	5,668,575,333
Male wings	MW1	55,372,384	85.86%	97.74%	94.75%	8,267,356,555
	MW2	36,205,752	85.60%	98.14%	96.47%	5,361,072,666
	MW3	38,739,102	86.18%	98.16%	96.50%	5,747,113,609

**Table 2 ijms-26-03562-t002:** Differentially expressed genes during *Gynaephora qinghaiensis* development.

Pairs	Up	Down	Total
FPW-vs-MPW	166	311	477
FW-vs-MW	4592	2935	7527
FPW-vs-FW	1474	1390	2864
MPW-vs-MW	1551	1754	3305

**Table 3 ijms-26-03562-t003:** Sequences of primers used for real-time fluorescent quantitative PCR.

Primer Name	Primer Sequence (5′–3′)	Product Size
*Burs*	F:TTGCCAAGAATCTGGTGAG	138
	R:TCCTCAATACCGCCACAT	100
*Ecr*	F:ACGAGGATTCCGACTTACCAT	149
	R:TTGGCGAATCCTTGTAGACCT	123
*Fringe*	F:GCAAGTGGTGGCAGATTCATCA	104
	R:GCCAGTAGCTTCATTTGTTCCA	104
*GST1*	F:CGACCAACGTCTATACTTCGACA	108
	R:AGAGCATCCTCGATCTTAGCTT	108
*GST2*	F:CACAACATACCATTCCGAT	116
	R:AAGAGAATCATCTTCACCGTA	127
*GST3*	F:GGCCGAGTCTCTATACCCT	116
	R:AGGCAACTTCTTTAACTCTGA	115
*GST4*	F:TGATTACGAACGTACCGCTGAC	122
	R:AAGTGGTCTGCTGCCGCATAA	93
*GST5*	F:GCAACCAACCTTCTCCTCGGAATAG	126
	R:AGCCGCTATGTACTCCTTCACTCTT	123
*GST6*	F:CGCTGGTGATAAGCTAACACT	149
	R:CCAGCTCGAACCATTTGACA	100
*Patched*	F:TCGATGCTGCTAGTATTCCC	108
	R:GCTCAGAAATCTTAGGTCGTT	138
*Rps6-1*	F:TACATAATGCTGTCCGGCAGA	100
	R:GACCGCCATATACCACTCGAAC	149
*Rps6-2*	F:AAGGTGCTGCCGCTGTTCGT	123
	R:ACATCATCTTCACTTGCCAGTCTAGGT	104
*Srf*	F:CACCGACTACCAGCGAGCA	104
	R:TTCATCTCCACTGTCGCCTGA	108
*Ubx*	F:CCAGTGCAGCATCAACCCA	108
	R:ATACGTTTGTCTTCCTCGTCT	116
*UGT1*	F:CTCTGTATGCCGATCAAGC	127
	R:GATCCGACAACAAATTACGAA	116
*UGT2*	F:AACTCCATACTCTTATGACGGATGAGC	115
	R:AGGATATGTTGGCTGTAGGTGACTTC	122
*UGT3*	F:GAAGTGCTTGCAGGCTTATCTGTAGTA	93
	R:AATGGCGGTATAGACGAAGACATGAC	126
*Vg*	F:GTGGTGTTCACGCACTACTCG	123
	R:CCATCGGCACACTCTCTTTAGG	149
*Rps15*	F:GTTGGCTCTCTATTGTAGGTATC R:AGGCTTGTATGTGACTGA	100

## Data Availability

The original contributions presented in the study are included in the article; further inquiry can be directed to the corresponding author.

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
