# Peer review of "Comparative Analysis of Transcriptome Data of Wings from Different Developmental Stages of the Gynaephora qinghaiensis"

_ijms, 2025, doi:10.3390/ijms26083562_

Round 1
Reviewer 1 Report
Comments and Suggestions for Authors
In the manuscript named “Comparative analysis of transcriptome data of wings from different developmental stages of the Gynaephora qinghaiensis”, Guixiang Kou et al have performed RNA-seq analysis to identify key genes involving in Gynaephora qinghaiensis wing development process, which would be helpful for exploring pest development and pest control works. However, there were some comments about it.
(1) The methods was not clearly described, for example, authors have described as “The reads obtained from the sequencing were compared with the Unigenes library” (line 437), but in results section, authors have described as “The sequences were aligned with the reference genome of G.qinghaiensis” (line), which was adopted by authors? Please clarify it.
(2) Which was “Mapping rate” in table 1, mapping to genome? Which was too high for RNA-seq analysis, please check it.
(3) The authors have described as “After de novo assembly, a total of 60 536 Unigenes were obtained”, which meant authors had performed de novo assembly works, but it was missing in method section. In addition, authors haven’t displayed this results in detail, while authors have annotated these unigenes by blast analysis, but the annotation rate was so low, about 41.6% (25162/60536). The results would indicate results was incredulous, please check it.
(4) Wayne diagram (figure 1)? Was it Venn plot?
(5) Was there some overlap between Figure 4 and Figure 5? Figure 4 shows the fold-change comparison between the two samples, while Figure 5 only showed expressional levels of the qRT-PCR results. Or were the samples in the two figures different or the same? Please check it. In addition, the RNA-seq and qRT-PCR would be performed correlation analysis, authors could provide correlation coefficient for each gene or all genes to illustrate reliability of results.
(6) The results were not well discussed in discussion section, there were many description words, such as “60 536 unigenes were identified from the transcriptome sequencing data, with an N50 length of 9342 bp and an average length of 1011 bp”, line 295. Even more, the “N50 length of 9342 bp” was error to their results, please check it.
(7) The writing of the conclusion section was also not standardized. It was recommended to concisely and accurately describe the important conclusions, rather than simply and repetitively listing how many genes there were in some comparisons.
Of all, there were many data or descriptions needed to be check, such as “Wayne diagram”, “N50 length of 9342 bp”, etc, they must be corrected before publish.
Author Response
please see file attached

Reviewer 2 Report
Comments and Suggestions for Authors
In this manuscript, the authors examined the transcriptomes of the developing pupal wings and the adult wings of Gynaephora qinghaiensis. They identified a number of differentially expressed genes. Overall, the transcriptomic analysis seems to be done well although I had some questions about how the wings develop and whether the appropriate time points were sampled for the assays. The manuscript is well written; however, please see my comments below.
Major comments
- One of the major issues I see is that it is not clear when the males and females begin to differ in their wing development. Is it during the pupal stage or earlier? I think this needs to be made clear in the introduction. It might be even better if you can provide photos of the female and male pupae as well as those of adults.
- Relatedly, I also was confused about the sample collection time. In the methods, the authors wrote “samples of qinghaiensis in two different developmental stages (pupal stage: 2nd instar; adult: 3rd instar) were selected for investigation.” How many larval instars do they have and when were the tissues collected? Typically there are several larval installs and we don't call the pupal stage and adults stage, 2nd and 3rd instars. Samples should have been collected when the wings were developing. Genes are turned on and off dynamically throughout development, so if you want to see what genes underly development of a particular structure, you want to collect tissues during the specification, determination and differentiation stages, not after the adult structures have developed. It looks like the authors examined gene expression in adult tissues. I am not surprised that the gene expression differed there but rather than ascribing the differential gene expression to development, I think it needs to be framed as tissue homeostasis or something like that since wing development has already terminated by then.
- Throughout the manuscript: nymph should be reserved for juveniles of hemimetabolous insects. For holometabolous insects, such as Lepidoptera, use pupae. It took me quite a while to figure out that the authors were talking about pupae, not larvae.
- Some of the results could probably be moved to the Appendix. For example, I am not quite sure what the significance is for section 2.2. Figure 1 could probably be moved to Supplemental materials.
- I am not sure it’s necessary to create a separate section for 3.6. It only has one sentence. It would be fine to just add it to the previous/next section.
- Please make sure all figures are referred to in the text.
- The Methods section is numbered oddly and placed in an odd location of the manuscript – I think the methods should appear either after the introduction or after the Conclusion section.
- Figure 2. I am not familiar with this depiction of GO terms. For readers who might be unfamiliar with it, I think it would be good to explain the figure. What do the number mean etc.
- Line 353: The authors wrote “Our experimental results showed that Ubx expression was downregulated between female nymphs and adult females, while showing upregulation in the other three groups. It is thus hypothesized that this gene is involved in the development of the anterior and posterior wings of male adults of Gynaephora qinghaiensis…” This is different from what is shown in Fig. 5 where FPW has the highest Ubx expression. I think this finding is rather interesting as Hox dosage has been shown to be responsible for the alteration of wing size: https://doi.org/10.1038/s41467-021-23293-8 Perhaps consider explaining how Ubx might contribute to the development of wing buds.
Minor comments
- Line 43, 44,111, 250, 303: Please italicize G. qinghaiensis
- Line 72: Please italicize Spodoptera litura
- Lines 81-86 is a run-on sentence
- Line 90: “Aphis citricidus,pest” sbould be “the pest, Aphis citricidus,”
- Line 105: Please italicize Tribolium castaneum
- Line 129: As it stands, FPW, FW etc have not been defined yet. These should be defined at the first mention in the text. Similarly, the gene names are not defined at first mention. If the methods section is moved up to right after the Introduction, this issue will be resolved.
- Line 254: Please italicize Vg
- Line 255-258: I think the directionality of expression change should be indicated rather than just saying the change in expression was significantly different (e.g. XX was expressed at significantly higher level compared to YY)
- There are two figure 3’s.
- Line 372: please italicize Srf
- Line 379: Dpp is not a hormone.
- Line 401: please italicize Fringe
Author Response
please see file attached

Round 2
Reviewer 1 Report
Comments and Suggestions for Authors
Thanks for authors’ works, the review had been well revised, most of my comments were well addressed in revision. However, there were some errors still in manuscript, for example, it was described as “were all above 95%” in line199, while it was about 82%-86% in table 2 (Mapping rate), please check it. The “Wayne diagram” was still in manuscript, but authors have claimed they revised it, please check it. In addition, I'm still concerned about the situation of the relatively low function annotation rate, or blastx mapping rate, see figure 1. Good luck.
Author Response
1.Comments: Thanks for authors’ works, the review had been well revised, most of my comments were well addressed in revision. However, there were some errors still in manuscript, for example, it was described as “were all above 95%” in line199, while it was about 82%-86% in table 2 (Mapping rate), please check it. The “Wayne diagram” was still in manuscript, but authors have claimed they revised it, please check it. In addition, I'm still concerned about the situation of the relatively low function annotation rate, or blastx mapping rate, see figure 1. Good luck.
Response: We have recorrected the Mapping rate and it should be above 80%. Venn diagrams are used for diagrams that show possible relationships between sets. In a Venn diagram, collections are usually represented by circles or oval regions, and the relationships between different collections are shown by the overlap of these regions. In this study we were able to find out the common genes and the number of genes annotated to between each database, and through the feasibility of these data support the results of the analysis later in 2.2. Therefore, we believe that the Wayne diagram should be placed in the text. We have supplemented 2.3 in Material Methods with sequence assembly and acquisition of complete unigenes. In a biological sense, a low unigene annotation rate does not mean that these genes are unimportant; many low-annotated unigenes may be expressed only in specific tissues, developmental stages, or environmental conditions, performing unique and as-yet-unrecognized functions, or it may be the case that they have developed unique sequences in evolution that have a low degree of similarity to genes with known functions. Therefore. Our transcriptome database can support subsequent data analysis.

Reviewer 2 Report
Comments and Suggestions for Authors
Thank you for your detailed responses. There are a number of things that remain unchanged in the manuscript.
1) In the introduction, please make sure to clarify that the wing degenerates during the pupal stage. This is critical information for the readers.
2) Please replace "nymph" with "pupa". Pupa is used in the methods and should be used throughout the manuscript instead of nymph.
3) L81 is still a run-on sentence.
4) In L119: Please replace (pupal stage: 2nd instar; adult stage: 3rd instar) with (pupal stage; adult stage)
5) Fig 4 and Fig 5 are not referred to in the results section. They need to be referenced there.
6) L91: the pest, should be unitalicized
7) L111 G. qinghaiensis needs to be italicized
Author Response
1.Comments:In the introduction, please make sure to clarify that the wing degenerates during the pupal stage. This is critical information for the readers.
Response: We can make sure that the wings of male and female adults become different from the pupal stage to the adult stage. The diagram below illustrates this.
2.Comments:Please replace "nymph" with "pupa". Pupa is used in the methods and should be used throughout the manuscript instead of nymph.
Response: We have replaced "nymph" with "pupa" throughout the text.
3.Comments:L81 is still a run-on sentence.
Response: We have corrected it.
4.Comments:In L119: Please replace (pupal stage: 2nd instar; adult stage: 3rd instar) with (pupal stage; adult stage)
Response: We have corrected it.
5.Comments:Fig 4 and Fig 5 are not referred to in the results section. They need to be referenced there.
Response: We have cited Figures 4 and 5 in the text.
6.Comments:L91: the pest, should be unitalicized
Response: We have corrected it.
7.Comments:L111 G. qinghaiensis needs to be italicized
Response: We have corrected it.
